# Therapeutic Strategies Targeting Tumor Suppressor Genes in Pancreatic Cancer

**DOI:** 10.3390/cancers13153920

**Published:** 2021-08-03

**Authors:** Kung-Kai Kuo, Pi-Jung Hsiao, Wen-Tsan Chang, Shih-Chang Chuang, Ya-Han Yang, Kenly Wuputra, Chia-Chen Ku, Jia-Bin Pan, Chia-Pei Li, Kohsuke Kato, Chung-Jung Liu, Deng-Chyang Wu, Kazunari K. Yokoyama

**Affiliations:** 1Division of General & Digestive Surgery, Department of Surgery, Kaohsiung Medical University Hospital, Kaohsiung 80756, Taiwan; kkkuo@kmu.edu.tw (K.-K.K.); wtchang@kmu.edu.tw (W.-T.C.); chuangsc@kmu.edu.tw (S.-C.C.); R030126@kmu.edu.tw (Y.-H.Y.); 2Regenerative Medicine and Cell Therapy Research Center, Kaohsiung Medical University, Kaohsiung 80708, Taiwan; r020017@kmu.edu.tw (K.W.); r991046@gap.kmu.edu.tw (C.-C.K.); r060139@gap.kmu.edu.tw (J.-B.P.); 1100058@kmuh.org.tw (C.-P.L.); 1020590@ms.kmuh.org.tw (C.-J.L.); dechwu@kmu.edu.tw (D.-C.W.); 3Department of Surgery, College of Medicine, Kaohsiung Medical University, Kaohsiung 80708, Taiwan; 4Department of Internal Medicine, Division of Endocrinology and Metabolism, EDA Hospital, College of Medicine, I-Shou University, Kaohsiung 82445, Taiwan; ed112609@edah.org.tw; 5Graduate Institute of Medicine, Kaohsiung Medical University, Kaohsiung 80708, Taiwan; 6Department of Infection Biology, Graduate School of Comprehensive Human Sciences, the University of Tsukuba, Tsukuba 305-8577, Japan; kkato@md.tsukuba.ac.jp; 7Division of Gastroenterology, Department of Internal Medicine, Kaohsiung Medical University Hospital, Kaohsiung 80756, Taiwan; 8Cell Therapy and Research Center, Kaohsiung Medical University Hospital, Kaohsiung 80756, Taiwan

**Keywords:** BRCA1, BRCA2, clinical trial, pancreatic cancer, p53, translational research, tumor suppressor gene

## Abstract

**Simple Summary:**

Tumor suppressor genes are critical in the control of many biological functions. They can be classified based on their roles in proliferation, cell-cycle progression, DNA repair/damage, and crucial signaling functions, including apoptosis, autophagy, and necrosis. The absence of functional tumor suppressor genes entails a higher risk of dysfunction of cell growth, differentiation, cell death, and cancer development. Loss of function or mutations of such genes has been identified in many types of cancer, such as breast, bladder, colorectal, head and neck, lung, ovarian, uterine, and pancreatic cancers. Familial cancer syndromes, such as Li–Fraumeni syndrome, are associated with loss of TP53 function. Extensive studies have been carried out to clarify the roles of the products of these genes, as well as their mechanistic link to cancers, to identify novel targets for specific cancer types. Here, we introduce the roles of tumor suppressor gene products in pancreatic cancer development and its therapeutics for tumorigenesis prevention.

**Abstract:**

The high mortality of pancreatic cancer is attributed to the insidious progression of this disease, which results in a delayed diagnosis and advanced disease stage at diagnosis. More than 35% of patients with pancreatic cancer are in stage III, whereas 50% are in stage IV at diagnosis. Thus, understanding the aggressive features of pancreatic cancer will contribute to the resolution of problems, such as its early recurrence, metastasis, and resistance to chemotherapy and radiotherapy. Therefore, new therapeutic strategies targeting tumor suppressor gene products may help prevent the progression of pancreatic cancer. In this review, we discuss several recent clinical trials of pancreatic cancer and recent studies reporting safe and effective treatment modalities for patients with advanced pancreatic cancer.

## 1. Introduction

Pancreatic ductal adenocarcinoma (PDAC) is a complex and a high-volume cancer that is managed with a multidisciplinary approach. The incidence of PDAC is gradually increasing worldwide, and its 5-year survival rate is about 10% [1,2]. The aggressive and lethal nature of this type of cancer is attributed to delayed diagnosis and lack of effective treatments. Despite the development by surgeons of many novel surgical techniques, such as superior mesenteric artery-first approach and superior mesenteric vein/portal vein resection and reconstruction, the silent nature of PDAC and its presentation leave a small percentage of patients qualifying for surgery (~20%). The most common therapy for PDAC is chemotherapy using modified FOLFIRINOX (mFOLFIRINOX) or gemcitabine-based regimen with the addition of capecitabine or nano-paclitaxel (Abraxane). Based on three randomized clinical trials (*n* = 2089), Galvano et al., concluded the optional adjuvant regimen for resected pancreatic cancers is mFOLFIRINOX. This robust scientific evidence strongly supports the pre-operative use of mFOLFIRINOX to increase the chance of R0 resection and reduce the incidence of micro metastases [3]. However, none of these regimen target altered genes. After neoadjuvant chemotherapy, surgical resection, and adjuvant chemotherapy with FOLFIRINOX (folinic acid, fluorouracil, irinotecan, and oxaliplatin), a small subgroup of patients can reach a mean survival of 54 months [4]. Pancreatic cancer is very aggressive and a prolonged treatment from diagnosis-to-initiation for PDAC may impact the survival. A study showed the optimal time was “within 6 weeks from diagnosis” when it was associated with an improved survival [5]. Nearly 80% of patients with PDAC cannot receive surgery at the time of diagnosis, and chemotherapy and radiotherapy do not have a significant impact on the overall survival of these patients. Recent advancements in the modality of irreversible electroporation have shown that it is safe and effective; however, it is not frequently used. Thus, new technologies are required that focus on target genes during the development of treatment algorithms for pancreatic cancer. Here we focus on tumor suppressor genes for new genetic trials for pancreatic cancer [6].

A genetically engineered mouse model of PDAC has successfully recapitulated human PDAC cancer biology [7,8,9]. This model was established using a combination of a mutant *Kras* oncogene with one deleted or mutated tumor suppressor gene (TSG, such as *p53*, *Smad4*, *p16^Ink4a^* (cdkn2a), or *Brca1/2*) [10,11,12]. The activated *Kras* oncogene can change the morphology of normal epithelial cells to the pancreatic intraepithelial neoplasm-1 (PanIN-1) type and initiate tumorigenesis [13,14,15]. In the later stage of cancer development, mutation/deletion of *p53* occurs, thus accelerating the disease and leading to its evolution to invasive, advanced PDAC [15].

*Kras* is a notorious oncogene for PDAC. About 95% of patients with PDAC carry mutant and activated *Kras*. *Kras* mutation occurs in the earliest precancerous lesions (such as PanIN-1) [13]. Furukawa et al. [16,17] proposed “the RAS–MAPK pathway with abrogation of dual specificity phosphatase 6 (DUSP6)” as the molecular mechanism underlying PDAC development (Figure 1).

Activation of Kras and inactivation of p16^Ink4a^ (CDKN2A) lead to the formation of low-grade PanIN-2 lesions [18]. Moreover, cell proliferation increases after *Kras* oncogene activation. To maintain the balance between cell division and apoptosis, TSGs are turned on and cell-cycle arrest is triggered. However, if TSGs are inactivated because of deletion or mutation, the increased cell cycles are no longer inhibited. Cell proliferation increases when the *Kras* oncogene is activated.

Additional loss of p53 and Smad4 functions will accelerate disease progression, and tumors become high-grade PanIN-3 lesions [7,19,20]. Finally, additional inactivation of DUSP6 results in advanced pancreatic ductal adenocarcinoma [16,17]. In this case, TSGs were found to help cellular DNA-repair homeostasis, control cell division, and induce apoptosis. p53, Smad4, p16, Brca1/2, and PTEN are common examples of TSGs that are involved in PDAC carcinogenesis [10,11,12,21,22,23] (Figure 2).

The next-generation sequencing (NGS) technique can identify a series of somatic mutations, such as fusions (ALK, NRG1, NTRK, and ROS1), mutations (*BRAF, BRCA1/2, HER2, KRAS,* and *PALB2* [24]), and mismatch repair (MMR) genes (*MLH1*, *MSH2*, *MSH6*, and *PMS2* [25]) in patients. These are mostly the products of tumor suppressor genes.

In the Ontario Pancreas Cancer Registry study [26], the germline DNA from 290 patients with varying degrees of family history was sequenced using a panel of 13 genes (*APC*, *ATM*, *BRCA1*, *BRCA2*, *CDKN2A*, *MLH1*, *MSH2*, *MSH6*, *PALB2*, *PMS2*, *PRSS1*, *STK11*, and *TP53*). Chaffee et al. [27,28] reported the results of sequencing using a panel of 25 cancer genes among 303 patients with a family history of pancreatic cancer. They found germline mutations in 10 genes (*ATM*, *BRCA1*, *BRCA2*, *CDKN2A*, *PALB2*, *PMS2*, *BARD1*, *CHEK2*, *MUTYH*, and *NBN*), which accounted for 11.6% of the prevalence of the overall PDAC cases.

Based on the data mentioned above, we understand that TSGs play important roles in PDAC carcinogenesis. In this review article, we attempt to summarize some of the potential treatments for these TSGs in PDAC.

## 2. BRCA1/2 Tumor Suppressor Gene—PARP Inhibitors

The loss of the wild-type allele of *BRCA*, which is considered a classical tumor suppressor gene, increases the risks of breast, ovarian, pancreatic, and prostate cancer, among others [29,30]. The frequency of *BRCA1/2* mutation among the whole PDAC population was estimated at around 4–7% [31,32]. Germline mutations mainly in the *BRCA2* gene lead to an increased risk of breast cancer, as well as a higher risk of developing PDAC, with a 2–6-fold increase in cancer risk compared with the general population.

The *BRCA* pathway, including *PALB2*, *FANCC*, and *FANCG*, involves the repair of DNA inter-strand cross-links. Patients with metastatic PDAC were initially frequently treated with platinum-based chemotherapy; currently, the FDA has approved olaparib, a poly (ADP-ribose) polymerase (PARP) inhibitor, as a maintenance regimen for adult patients with PDAC and deleterious germline *BRCA* mutations. Olaparib can trap the PARP-1 protein at a single-strand break/DNA lesion and disrupt its catalytic cycle, ultimately leading to replication fork progression and consequent double-strand breaks (Figure 3) [33,34]. The efficacy of this regimen was documented in the POLO (NCT 02184195) study [33], which reported a median progression-free survival (PFS) of 7.4 months (95% CI: 4.1, 11.0) among patients who received olaparib compared with 3.8 months (95% CI: 3.5, 4.9) for patients who received the placebo (HR 0.53; 95% CI: 0.35, 0.81; *p* = 0.0035).

### 2.1. Clinical Trials of Inhibitors of PDAC

The recent clinical trials of inhibitors of human pancreatic cancers are summarized as below (Table 1, Table 2, Table 3 and Table 4). We showed the summary of the inhibitors of MDM2, PARP, and gene-based target therapy drugs (Table 3) and immunotherapy drugs (Table 4).

### 2.2. Other PARP Inhibitors for the Treatment of Pancreatic Cancer

Similar drugs that function based on the “synthetic lethality” concept are indicated below.

#### 2.2.1. Veliparib

In a phase 1 clinical trial (NCT01908478), dose-escalated veliparib was used in 30 patients with locally advanced or borderline resectable pancreatic cancer, which was combined with weekly gemcitabine treatment and daily radiotherapy. This study confirmed that veliparib is safe and well tolerated in combination therapy with gemcitabine and radiotherapy among patients with PDAC [34].

#### 2.2.2. Talazoparib (MDV3800 or BMN 673)

A novel inhibitor of PARP was developed that is more potent than the previous PARP-1/2 inhibitors. A phase 1 study (NCT01286987) confirmed the antitumor activity and maximal tolerable dose (1.0 mg/day) of talazoparib. Four of the 13 patients with PDAC included in the trial showed clinical benefits (rate, ~31%, ≥16 weeks) [35].

#### 2.2.3. Rucaparib (NCT02042378)

A phase 2 study was carried out to measure the efficacy and safety of rucaparib in patients with BRCA1/2 mutations. Patients with PDAC with measurable locally advanced/metastatic lesions were enrolled in this trial. Nineteen subjects (16 had germline mutations and three had somatic mutations) received oral rucaparib (600 mg twice daily) after the administration of one-to-two prior chemotherapy regimens. Two partial responses and one complete response (CR) were confirmed (objective response rate, 15.8%; 3 out of 19 cases). The disease control rate (CR, partial response, or stable disease for ≥12 weeks) was 31.6% (6 out of 19 cases). This study provided evidence that rucaparib has an acceptable safety profile and is beneficial for patients with advanced PDAC [36].

At present these inhibitors are used for clinical trials for PDACs. Moreover, some of them are also useful for the treatment of breast cancers. Because BRCA1/2 mutations and the PARP activities are closely related to each other, PARP inhibitors are useful for the treatment of PDACs. Trials in combination with other treatments should be explored further.

## 3. TP53 Tumor Suppressor Gene

There is no doubt that the *TP53* tumor suppressor gene is one of the most important genes in many cancers. It is mutated in nearly 50% of human cancers, including PDAC (mutation frequency, 58.7%), esophageal squamous cell carcinoma (93.7%), invasive breast cancer (32.7%), and non-small-cell lung cancer (66.5%) [37,38]. Patients with Li–Fraumeni syndrome who carry *TP**53* gene mutations in germline cells may have various cancers during their lifetime, with onset of the cancer at a younger age compared with the average patients [39,40]. However, the majority of *TP53* gene dysfunctions in sporadic cancers are point or missense mutations triggered by UV light, aflatoxins, smoking, or other environmental factors. Because of the high frequency of TP53 mutation in many types of cancers, therapeutic strategies targeting mutant TP53 have attracted great interest [38].

### 3.1. Gain-of-Function Mutant p53 (mutp53)

Mutant p53 proteins, which cannot activate the mouse double minute 2 homolog (MDM2), prolong the half-life of, and become gain-of-function (GOF) molecules in mutp53-harboring cancer cells. These GOF activities have tremendous effects on many important pathways, such as metastasis, proliferation rate, apoptosis/drug resistance, stem cells, chronic inflammation, genomic instability, and metabolism (see Figure 4 and Table 5). These mutp53 proteins function by reversing the effects of each fundamental reaction of cells, to inactivate cell proliferation, differentiation, reprogramming, stem cell function, cell cycle, and apoptosis, as well as metabolic control.

Donehower et al., reported that GOF mutp53 proteins can activate ~500 downstream genes involving several pathways that mainly regulate cell-cycle arrest, apoptosis, senescence, DNA repair, and genetic stability [41]. GOF mutp53 proteins interact with NF-kB, HIF-1alpha, SP1, Twist1, E2F, or SREBP1 via protein–protein interactions, to inhibit p63 and p73 activities. Di Agostino et al., demonstrated that the mutp53/NF-Y protein complex can aberrantly recruit p300 instead of histone deacetylases, eventually leading to an opposite cell-fate outcome [42]. GOF mutp53 can render cancer cells more aggressive, with aneuploidy, higher proliferation rate, and cancer stem cell phenotypes, such as stem cell markers and resistance to apoptosis and therapeutic drugs [43]. In addition, mutp53 contributes to chronic inflammation and angiogenesis in the cancer cell microenvironment, and eventually enhances distal metastasis. For example, 80% of cases of basal/triple-negative breast cancer have p53 mutations [44]. GOF mutp53 proteins reprogram metabolic and anabolic pathways and enhance the Warburg effect by increasing glucose uptake and lactate formation, despite the presence of functioning mitochondria and oxygen.

Because p53 mutation is one of the most common genetic alterations in cancers, approaches aimed at restoring “the tumor-suppressive function of wild-type p53 (wtp53)” are urgent and necessary. In turn, this can enhance chemo-radiotherapy sensitivities in mutp53-harboring cancer cells. However, the development of a strategy that effectively targets the GOF mutp53 proteins is challenging and difficult, mainly because a vast number of genes and pathways are altered by these proteins.

#### 3.1.1. Hotspots of mutp53

The six most common mutation hotspots of mutp53 have been identified, including the R175H, G245S, R248Q, R248W, R273H, and R282H residues. These hotspots account for over 28% of the total p53 mutations in various cancers [45]. GOF mutp53 proteins exhibit an altered structural conformation (structural mutations, such as R175H) or reduced capacity of binding to DNA (contact mutations, such as R273H) [46,47]. GOF mutp53 proteins not only lose their original wild-type tumor-suppressive function, but also accumulate to very high levels and exhibit GOF activities in mutp53-harboring cancer cells.

#### 3.1.2. Restoring wtp53 Function

Several small molecules can bind to mutp53 proteins and change their structural conformation to allow binding to DNA in the same manner as does the wtp53 protein (i.e., they can restore wtp53 protein function).

##### CP-31398

This synthetic small-molecule compound can restore the wtp53 transcription function and induce apoptosis in cancer cells by stabilizing the mutant protein. However, its mechanisms are controversial [48,49].

##### STIMA-1

Zache et al. [50] found that this low-molecular-weight compound has some structural similarities to CP-31398, stimulates mutant p53 DNA binding in vitro, induces the expression of p53 target proteins, and triggers apoptosis in mutant-p53-harboring tumor cells.

##### PRIMA-1 and APR-246

PRIMA-1 and its analog PRIMA-1MET, now termed APR-246 [51], can reactivate mutp53 and induce a wtp53 biological response, such as apoptosis, in tumor cells, thus inhibiting tumor growth in mice. Both PRIMA-1 and APR-246 have been tested in clinical trials that included patients with hematological malignancies or hormone-refractory prostate cancer (www.clinicaltrials.gov; NCT03268382, NCT03931291, NCT04214860) [52].

##### p53R3

p53R3 is a p53 rescue compound that inhibits the proliferation of cancer cells expressing mutp53 by inducing the expression of p53 target genes, including *p21^Cip1^*, *PUMA*, and *BAX*, to induce cell-cycle arrest and apoptosis in cancer cells [53].

##### PK083 and PK7088

PK083 and PK7088 were designed to bind the cavity created by mutp53^Y220C^. The Y220C mutation of p53 is an excellent “druggable” target [54,55]. Both PK083 and PK7088 bind to the Y220C mutant, restore the wtp53 conformation, and induce Y220C-dependent cell-cycle arrest and apoptosis [56,57].

##### RITA

RITA is another compound that can reactivate several mutant p53 proteins, such as those carrying the R175H, R248W, and R273H mutations [58].

##### Chetomin

Chetomin is a small molecule that can specifically reactivate the mutp53^R175H^ protein to the wild-type p53 conformation by increasing its binding capacity through HSP40. This drug selectively inhibits the growth of cancer cells harboring mutp53^R175H^, but not of those carrying mutp53^R273H^ [59]. Furthermore, it can also enhance the radiosensitivity of cancer cells, regardless of p53 status. The underlying mechanism occurs via interference with the hypoxia-inducible factor (HIF) pathway [60,61,62].

##### Phenethyl Isothiocyanate

Phenethyl isothiocyanate (PEI), which is a natural dietary-related compound that is present in cruciferous vegetables, can restore the wild-type conformation and transcriptional activity of mutp53^R175H^, sensitize mutp53^R175H^ to proteasomal degradation, and have a growth inhibitory effect on cancer cells expressing mutp53^R175H^. Dietary supplementation with PEI led to the reactivation of wtp53 activity in vivo and the inhibition of tumor growth in a xenograft mouse model. This represents the first example of mutant p53 reactivation by a dietary compound and may have important implications for cancer prevention and therapy [63].

#### 3.1.3. Zinc-Based Therapy

Zinc is a known regulator of p53 that is essential for its correct binding to target genes. Adding zinc to the mutp53 protein can reduce the effect of substitutions (G245C/G245D) on conformational changes [64]. The combination of adriamycin with zinc inhibited tumor growth in the transgenic MMTI-neu murine breast cancer model [65].

##### NSC319726/ZMC1

NSC319726/ZMC1 activates the allele-specific mutp53^R175^ and restores its wild-type structure and function [66]. The p53^R175^ mutation is the third most frequent missense mutation of this protein. The potential number of patients who may benefit from this compound is estimated at 32,000 per year in the United States [67]. NSC319726, as a p53^R175^ mutant reactivator, does not bind to mutp53; rather, it increases the intracellular zinc concentration and enhances the folding of R175H mutants.

##### COTI-2

COTI-2, which is a third-generation thiosemicarbazone compound, can chelate zinc ions and reactivate mutant p53 to its wild-type form. COTI-2 was shown to restore the normal DNA-binding properties of the p53 mutant protein through p53-dependent and -independent mechanisms [68].

It is evident that p53 mutation is one of the fundamental genetic mutations that trigger cancers. Even hemi-allelic mutation initiates the cancer program together with other critical mutations of cancer genes. Thus, recovering the mutation phenotypes to the normal wtp53 presentation is the main target in the prevention of p53-dependent cancers. Further trials targeting GOF mutp53 proteins are required.

## 4. GOF mutTP53 Proteins and Cancer Stem Cell Phenotypes

GOF of mutp53 proteins have pinpointed the critical steps that are involved in cancer-stem-cell-related genes and chemoresistance.

GOF mutp53 proteins involve a broad spectrum of mechanisms of chemoradio resistance, including resistance to apoptosis, autophagy inhibition, metabolic reprogramming, and increased expression of drug efflux pumps [66]. GOF mutp53 proteins favor self-renewal pathways by increasing EZH2 and YAP/TAZ activity and enhancing the surface expression of cancer stem cell markers, such as CD44, CD133, LGR5, and ALDH [69]. All these factors are important for cancer stem cells (CSCs), which are believed to be the origin of many cancers. Thus, GOF mutp53 plays an important role in CSC formation. Hassen et al., showed that Kras and the mutated GOF p53 are the main drivers of PDAC aggressiveness [70]. Mutant p53 and CREB1 upregulate the FOXA1 transcription factor and promote Wnt–β-catenin signaling to drive tumor metastasis [71]. Capaci et al., reported that the mutp53/HIF1α/miR-30d axis can potentiate the release of the soluble extracellular secretome to remodel the extracellular matrix, thus favoring cancer cell growth and metastatic colonization [72].

Thus, the interrelationship between the signaling of GOF mutp53 and chromatin modifiers, such as PRC2 and EZH2, and the molecular mechanisms underlying the generation of the cancer stem cell markers CD44, CD133, LGR5, and ALDH, should be clarified, and other signaling pathways, such as Wnt–β-catenin and FOXA1 should be elucidated. These molecular signaling pathways of CSCs are also the targets of future clinical research.

## 5. Targeting Mutant p53 Protein Stability

Mutant *p53* genes are pro-oncogenic drivers [73,74,75], and cancer cells rely on these oncogenes for survival or growth [76]. Reducing mutp53 protein stability leads to their degradation, which in turn leads to cancer cell death. Heat-shock proteins, such as HSP70 and HSP90, can bind to mutp53 proteins to form a chaperone complex. This complex can inactivate endogenous MDM2 and the carboxy terminus of the HSP70-interacting protein (CHIP), resulting in mutp53 protein accumulation in human cancer cells. The pharmacological inhibitor 17-allylamino-17-demethoxygeldanamycin (17AAG) can destroy this chaperone complex, liberate mutp53 proteins, and reactivate endogenous MDM2 and CHIP, to degrade mutp53 proteins [77] (Figure 5).

### 5.1. Geldanamycin

Geldanamycin, which is a highly specific HSP90 inhibitor, decreases the intracellular levels of mutp53 proteins via MDM2 degradation [78]. Because HSPs also regulate wtp53, HSP inhibitors have not received FDA approval for entering clinical trials.

### 5.2. Ganetespib

Alexandrova et al. [79] reported that ganetespib, which is a potent HSP90 inhibitor, can significantly extend the lifespan of homozygous mutp53^R172H^ mice.

### 5.3. Alvespimycin Plus SAHA

Similarly, a combination of alvespimycin (17DMAG) and SAHA (histone deacetylase inhibitor) effectively induces mutp53 protein degradation and tumor necrosis and prolongs the lifespan of mutp53^R172H^ mice [79].

The stability of mutp53 proteins is crucial for the progression of many cancers including PDACs. HSPs, MDM2/4, and the CHIP chaperone are among these regulators and might be possible targets for anti-cancer therapeutics.

## 6. Synthetic Lethality of p53 Loss

Mutp53-harboring cancer cells usually lack G1/S checkpoints and become more dependent on G2/M checkpoints to survive after DNA damage. Therefore, inhibition of G2/M checkpoint regulators, such as CHK1/2 and WEE1, has been reported to induce a mitotic catastrophe and synthetic lethality in mutp53-harboring cancer cells. ARK/CHK1 inhibitors, such as UCN-01, PF477736, and AZD-7762, may potentiate the cytotoxicity of chemoradio-genotoxic effects in mutp53-harboring human cancers. More than 70 clinical trials (various chemoradiotherapy regimens combined with different ARK/CHK1 inhibitors) have been carried out and reviewed in this context [80]. A combination of gemcitabine or olaparib (PARP1 inhibitor) with a potent WEE1 inhibitor (AZD1775) yielded synergistic lethal effects on ovarian and endometrial mutp53-harboring cancer cells [81]. A phase II clinical trial has proven that AZD1775 enhances carboplatin efficacy in patients with mutp53 ovarian cancer who were refractory to first-line platinum-based therapy [82]. Recently, Hartman et al., reported the in vivo and in vitro anti-tumor effects of the combination of AZD1775 with irinotecan or capecitabine/5-FU in patients with PDAC using a patient-derived xenograft model [83]. Xiao et al., combined a chemotherapeutic agent with a PARP inhibitor and showed their synergistic cytotoxicity in GOF mutp53 (particularly mutp53^R273H^) breast cancer cells [84]. Wei et al., also reported recently that a combination of gemcitabine with a protein arginine methyltransferase 5 (PRMT5) inhibitor yielded a synergistic lethal effect on a patient-derived xenograft PDAC model [85]. Thus, mutp53 and G2/M checkpoints for mitotic inhibition and the synthetic lethality of p53 loss are crucial for understanding cancer development.

## 7. Immunoregulation of the Microenvironment of PDAC

### 7.1. PD-1/PD-L1 in PDAC

Previous studies showed that immunotherapy with PD-1/PD-L1 failed to exert any effects on PDAC. The microenvironment of PDAC is not immunogenic or immunosuppressive [86]. In general, pancreatic cancer produces a local and systemic immune dysfunction or immunosuppression to avoid recognition and attachment by effective immune-competent cells. The microenvironment of cancers exhibits a lower number of tumor-infiltrating lymphocytes, and dendritic cells and many suppressor T cells [87]. Tumor cells use mechanisms that act via the PD-L1 or CTLA-4 program, a blockage of co-stimulation to activate T cells, and the recruitment of tumor-associated macrophages and marrow-derived suppressor cells (MDSCs), to achieve immune suppression [88]. Recently, Amin et al., reported that the addition of immunotherapies to adjuvant chemotherapy improved survival compared with chemotherapy alone after curative-intent resection of PDAC [89]. Cao et al., reported that p53-mediated PD-1 activation is involved in tumor suppression in an immunity-independent manner [90].

### 7.2. Mutant p53 GOF Mechanisms via the Shedding of the Tumor-Promoting Secretome (Including Exosomes)

Mutp53-harboring cancer cells secrete numerous extracellular factors (secretomes) to create a supportive microenvironment for their progression. These factors can be either soluble or contained within vesicles (such as exosomes). Cooks et al., reported that a mutp53-bearing colon cancer secreted exosomes enriched in miR-1246. These exosomes were taken up by nearby macrophages and reprogrammed them into a tumor-promoting M2 status [91]. Tran et al., isolated tumor-infiltrating lymphocytes (TILs) from a patient with metastatic cholangiocarcinoma containing CD4^+^ helper T cells that could specifically recognize a patient-specific mutant protein, the erbb2-interacting protein (ERBB2IP^E805G^). This immune response was presented by HLA-DQB1*0601, and the minimal neoepitope was located within the following 13-amino-acid sequence: NSKEETGHLENGN (where E is Glu; G, Gly; H, His; K, Lys; L, Leu; N, Asn; S, Ser; and T, Thr). The patient received two courses of this patient-mutant-specific helper-T cell therapy and exhibited tumor shrinkage. Three lung metastases, which were resected nearly a half year after this immunotherapy, were infiltrated by the ERBB2IP mutation-reactive T cells, suggesting that they contributed to cancer regression and stabilization of disease [92].

### 7.3. Neoepitopes from mutp53 Proteins Are Recognized by TCRs on CD8^+^ T Cells

To date, no ideal tumor-associated antigens have been identified, either localized within cells or mounted on the surfaces of normal cells. Small peptides (neoantigens) degraded from the mutp53 protein are secreted (as secretomes) from cancer cells, to be taken up by surrounding T cells. Previous experiments have shown that these small mutant peptides (neoantigens, about 10–15-peptide-containing mutated neoepitopes) are highly immunogenic and trigger the killing of cancer cells via both CD8^+^ and CD4^+^ immune responses [91]. These small peptides (neoantigens) are taken up by antigen-presenting cells. and formed an HLA/MHC1-peptide complex that is recognized by T-cell receptors (TCRs) expressed on the surface of CD8^+^ T cells. Interestingly, small peptides derived from hot-spot-mutp53 proteins can become “public neoantigens” [91]; i.e., hotspots carrying mutp53 genes and encoding these hotspot-mutant peptides frequently occur in many human cancers. These neoepitopes derived from hot-spot-mutp53 proteins can become “public” neoantigens [93].

### 7.4. CAR-T Cells Promote T-Cell Expansion to Promote Anti-Tumor Function

A comprehensive picture of the GOF of p53 is necessary to achieve personalized cancer treatment. Using the NGS method, physicians can now determine the landscape of the whole-genome data of patients. Pavlakis and Stieve [94] proposed that mutp53-harboring cancer cells secrete numerous secretomes (including miRNA-enriched exosomes) to recruit different immune/stellate cells and remodel the microenvironment to favor tumor promotion. In contrast, these recruited inflammatory and immune cells can also affect and shape the GOF activities of mutant p53 within cancer cells [95]. In the past decade, adoptive T-cell therapies (ACTs) have yielded excellent results for chronic lymphocytic leukemia [96]. Those authors isolated endogenous TILs for ex vivo expansion, genetically engineered TCRs, or chimeric antigen receptors (CARs) against cancer-specific antigens. Fifth-generation CAR-T cells were designed to activate the JAK–STAT pathway and promote T-cell expansion, which afforded a better antitumor effect [97]. A phase 1 clinical trial has demonstrated that anti-EGFR CAR-T treatment is safe and effective in patients with metastatic PDAC (median overall survival, 4.9 months; range, 2.9–30 months) [98]. The potential obstacles for ACT in PDAC include appropriate ways to deliver CAR-T cells to fibrotic, immunosuppressive cancer environments. However, whether mutp53 neoantigen-specific T cells can cause solid tumor regression has yet to be determined. These trials of cancer immunotherapies are also critical for developing new therapeutics for PDACs.

## 8. MDM2–MDMX(MDM4)–p53 Axis

Normal cells contain a lower level of p53 proteins because of their short half-life, which is modulated by a ring finger E3 ubiquitin ligase termed MDM2. Moreover, MDMX(MDM4), an analog of MDM2, enhances the activity of MDM2 further to polyubiquitinate p53 by forming a complex with MDM2 [99]. The mutual dependence of MDM2 and MDMX regarding their p53-inactivation function, as well as their essential roles in controlling p53 levels and activity in vivo, have been reported [100]. The target genes under the control of the MDM2–MDMX–p53 loop are primarily critical for carcinogenesis. Recently, small molecules, such as protoporphyrin IX (PpIX), have been used to target the p53/MDM2 and p53/MDMX interactions and promote alternatives to target wt-p53-bearing tumors, such as pancreatic carcinoma [101]. Therefore, the MDM2–MDMX–p53 pathway is subjected to multiple layers of control in response to various stress signals and cancers, including PDACs.

## 9. Future Perspectives

One of the possible interesting technologies for the treatment of solid tumors, including PDACs, is the liquid biopsy (LB) to obtain information for diagnostic, prognostic, and predictive purposes in the near future [102,103]. However, to date, this approach has been used in breast cancer, colorectal cancer, and melanoma exclusively. This new technique is emerging as an identification clinical tool. In current clinical practice, LB is used for the identification of driver mutations in the circulating DNA derived from both tumors and circulating neoplastic cells. Liquid derivatives other than LBs, such as circulating tumor cells, circulating tumor RNA/DNA, microRNA, platelets, and extracellular vesicles, as well as other biofluids, such as urine and cerebrospinal fluid, may be adopted to detect mutations of tumor suppressor genes and their related genes in the future.

## 10. Conclusions

This review focused on novel therapies to treat patients carrying mutated TSGs. However, the success of these treatments remains low. Collisson et al. [104] showed that there is a human QM-PDAC (quasi-mesenchymal) subtype in which the genetic alterations were similar to those of cell lines obtained from genetically engineered *p53*^Lox/+^ mice (lacking the tumor suppressor *p53* gene). Bailey et al., reported that the “squamous” subtypes of PDAC are more aggressive, metastatic, and undifferentiated tumors, because enriched *TP53* mutations were detected in this subtype [105].

The laser-capture microdissection technique was used to dissect purified PDAC cancer tissues, to study whole genomes and perform transcriptome analysis. Chan-Seng-Yue et al. [106] proposed two subtypes: the classical and basal subtypes. The basal-like-A phenotype (high squamous signature) is linked to major K-ras imbalances in late-stage disease. Conversely, the classical subtype is believed to have a better response to a chemotherapy regimen (mFOLFIRINOX), as well as a better prognosis. It is currently undergoing a clinical trial known as NCT04683315.

Genetic studies of patients with breast cancer yielded similar results. Most mutTP53 clusters were detected in the basal-like subtype of breast cancer, which is chemoresistant and has the worst prognosis [107]. Mutant p53 activates lipid metabolism in tumors; moreover, it binds to and activates a series of transcription factors, the sterol regulatory element-binding proteins, and induces the expression of many genes in the mevalonate pathway, which leads to the disruption of breast cancer cell architecture in a 3D culture model and contributes to increased proliferation, survival, invasion, and metastasis [107].

p53 is the master regulator of tumor suppressor genes, and the GOF mutp53 functions not only to generate cancer cells, but also to promote a higher proliferation capability by triggering EMT and promoting chemoradio resistance. Furthermore, its secretome remodels the extracellular matrix and creates an immune-suppressive microenvironment that promotes the migration and metastasis of cancer cells. A subtype of cancer cells acquire the GOF mutp53-specific characteristics of “cancer stem cells,” which is thought to be responsible for carcinogenesis and chemoradioresistance. Therefore, approaches aimed at eradicating these frequently encountered “mut-*TP53*-harboring cancer cells” are an important clinical issue for future studies. Understanding these issues is crucial for improving the current cancer treatments.

## Figures and Tables

**Figure 1 cancers-13-03920-f001:**
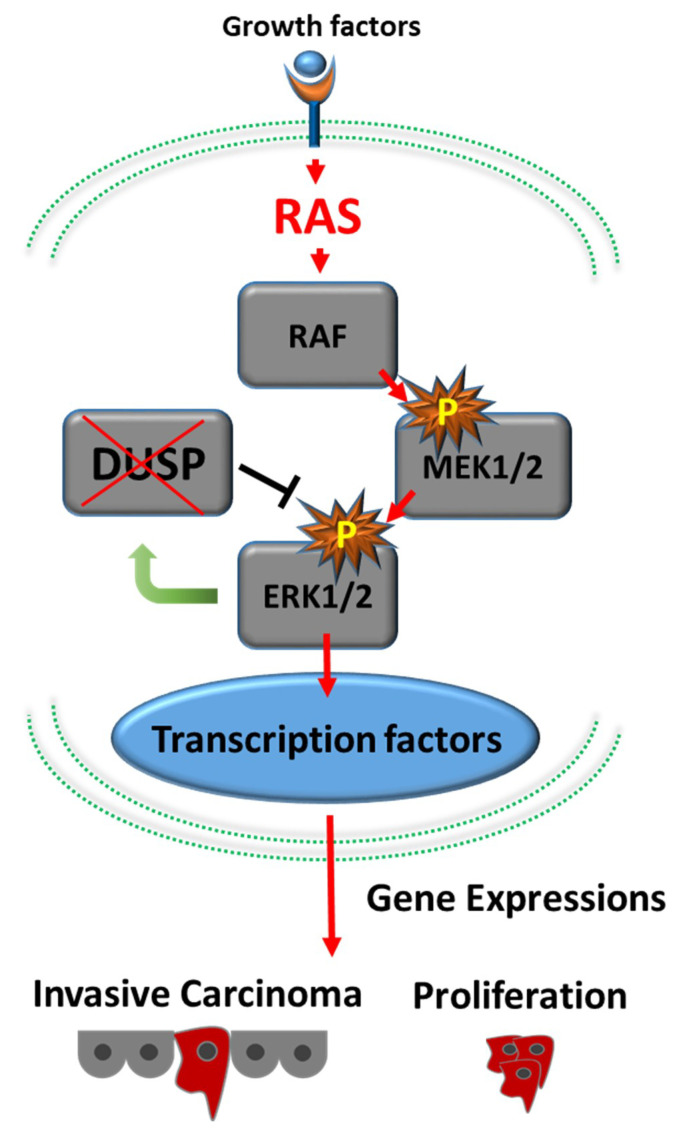
Development of pancreatic adenocarcinoma via the RAS–MAPK signaling cascade. Active RAS generated by mutated Kras activates downstream cascades, including RAF1–MAP2K1–MAPK1. Loss of expression of DUSP6 results in abrogation of the feedback loop between MAPK1 and DUSP6 and leads to constitutive activation of MAPK1, which eventually results in invasive phenotypes.

**Figure 2 cancers-13-03920-f002:**
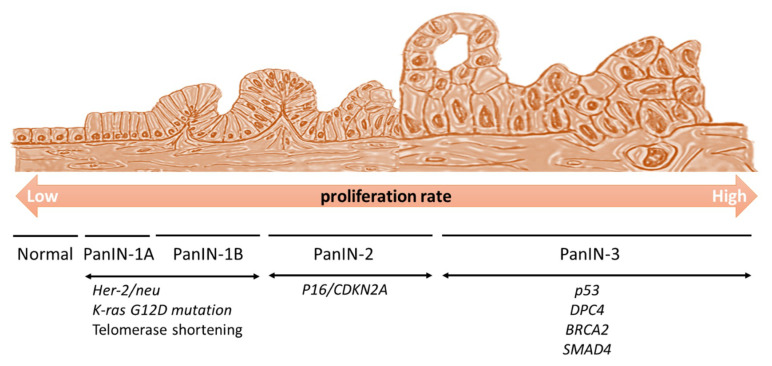
Generation of pancreatic intraepithelial neoplasia (PanIN). The *Kras* oncogene is activated and increases cell proliferation. Her-2/neu expression and telomere shortening also occur at the initial stages of the disease. If tumor suppressor genes are inactivated, there is no stopper, and the cell cycle proceeds extensively. p16^Ink4a^ is activated at the PanIN-2 stage, and then p53, DPC4, and BRACA2 are activated at the PanIN-3 stage, to induce a high cellular proliferation. Subsequently, the cells enter the neoplastic stage.

**Figure 3 cancers-13-03920-f003:**
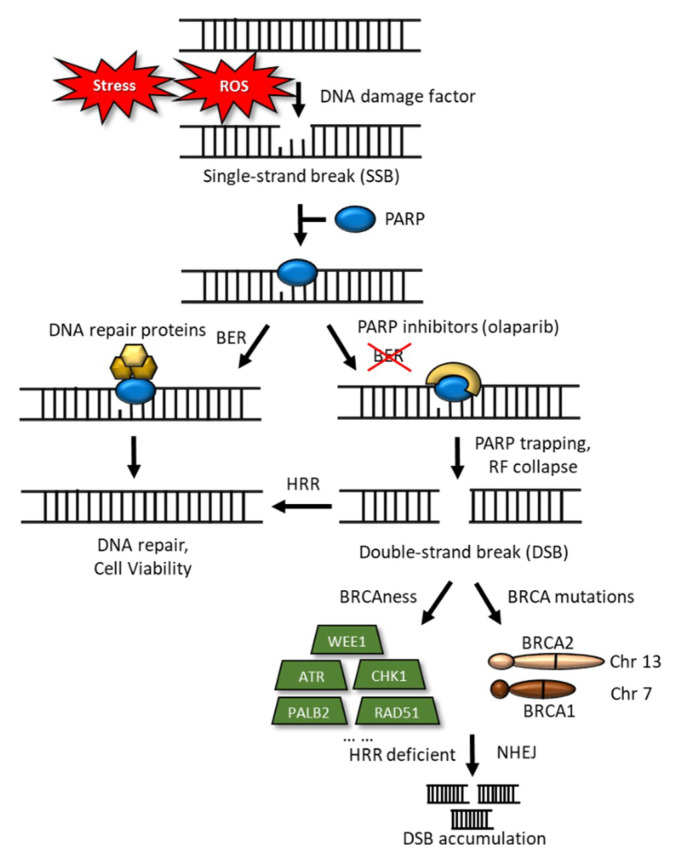
Schematic representation of DNA damage and DNA-double-strand breaks. DNA-damage inducers, such as stress and ROS, sometimes trigger single-strand breaks (SSBs), to which PARP binds, for their repair. Base excision repair (BER) reverses the DNA damage resulting from oxidation, deamination, and alkylation. In this case, BER DNA glycosylase recognizes and removes the damaged base, leaving an abasic site that is processed further, by short-patch or long-patch repair, which largely uses different proteins to complete BER. PARP inhibitors trap the PARP-1 protein at an SSB/DNA lesion and disrupt its catalytic cycle, this would ultimately lead to replication fork progression and consequent double-strand breaks (DSBs). In the case of *BRCA* mutations, loss of HRR would result in cell death. HRR deficient and nonhomologous end-joining lead to the accumulation of DSBs.

**Figure 4 cancers-13-03920-f004:**
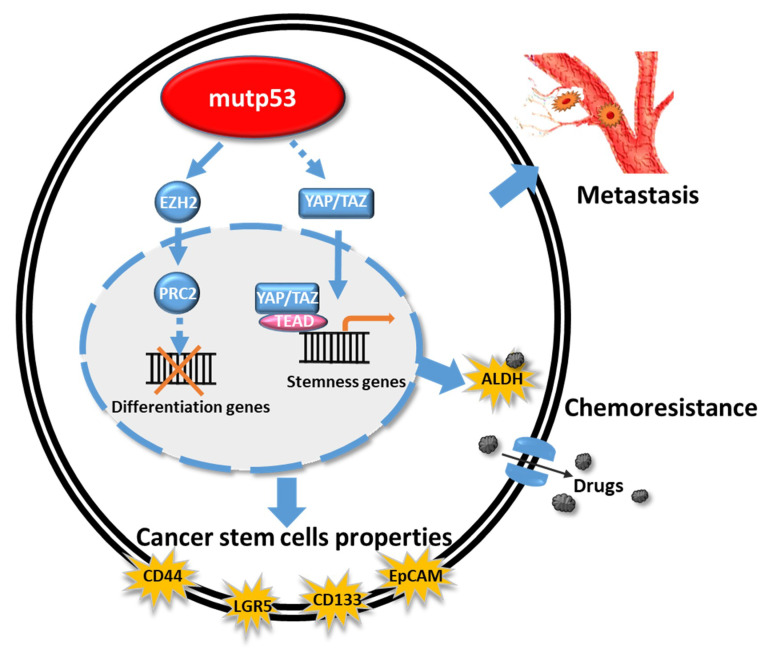
Schematic representation of the effects of mutp53 on cancer-stem-cell–related genes and chemoresistance function. Mutp53 induces EZH2 to block the function of differentiation genes and triggers YAP/TAZ signaling to upregulate stemness genes, which are induced to chemoresistance-related genes. Mutp53 also induces the cancer stem cell related genes, such as CD44, LGR5, CD133, and EpCAM, which might induce the metastasis and cancer progression.

**Figure 5 cancers-13-03920-f005:**
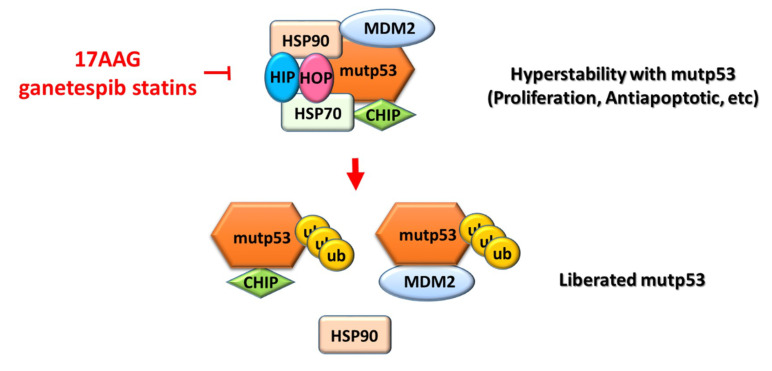
Schematic modeling of the modulation of mutp53 stability by the HSP90 multichaperone machinery in cancer cells. Heat-shock proteins (such as HSP70 and HSP90) bind to mutp53 proteins to form a chaperone complex [77]. This complex can be inactivated by endogenous MDM2 and the carboxy terminus of the HSP70-interacting protein (CHIP), resulting in the accumulation of mutp53 proteins in human cancer cells. The pharmacological inhibitor 17-allylamino-17-demethoxygeldanamycin (17AAG) can destroy this chaperone complex, liberate mutp53 proteins, and reactivate endogenous MDM2 and CHIP, to degrade mutp53 proteins [77]. CHIP, carboxy terminus of the Hsc70-interacting protein; HIP, heat-shock protein (HSP) 70 co-chaperone; HOP, HSP70/90 organizing protein; Ub, ubiquitinated proteins.

**Table 1 cancers-13-03920-t001:** Recent clinical trials of MDM2 inhibitors in human pancreatic cancers.

Trial ID	Therapeutic Drug	Phase	Status	Condition	Primary Outcome
NCT03654716	Drug: ALRN-6924 Drug: Cytarabine	Phase 1	Recruiting	Solid Tumor	Percentage of patients with dose limiting toxicity by CTCAE V.5.0 for each dose level (Time Frame: 2 years) Percentage of patients with toxicity by CTCAE V.5.0 (Time Frame: 2 Years)
NCT02098967	Drug: RO6839921	Phase 1	Completed	Neoplasms	Incidence of adverse events (Time Frame: Approximately 1 year) Incidence of dose-limiting toxicities (Time Frame: Approximately 1 year)
NCT01462175	Drug: RO5503781	Phase 1	Completed	Neoplasms	Maximum Tolerated Dose (MTD) (Time Frame: Up to 28 days) Percentage of Participants With Dose Limiting Toxicities (DLTs) (Time Frame: Up to 28 days) Percentage of Participants With Adverse Events (AEs) and Serious Adverse Events (SAEs) (Time Frame: approximately 1.5 years)
NCT03362723	Drug: Idasanutlin	Phase 1	Completed	Solid Tumors	Area Under the Curve (AUC) of Idasanutlin ( Time Frame: Predose (within 2 h), 1, 2, 4, 6, and 10 h on Days 1, 8, 15, and 22; on Days 2, 3, 5, 9, 10, 12, 16, 17, 19, 23, 24, and 26; Day 29 (end of Cycle 1) or Cycle 2 Day 1 (for participants in the optional treatment extension phase) (cycle = 28 days)) Maximum Observed Plasma Concentration (Cmax) of Idasanutlin (Time Frame: Predose (within 2 h), 1, 2, 4, 6, and 10 h on Days 1, 8, 15, and 22; on Days 2, 3, 5, 9, 10, 12, 16, 17, 19, 23, 24, and 26; Day 29 (end of Cycle 1) or Cycle 2 Day 1 (for participants in the optional treatment extension phase) (cycle = 28 days))
NCT03714958	Drug: HDM201 Drug: Trametinib	Phase 1	Recruiting	Advanced Cancer Metastatic Cancer	Dose Maximum Tolerated (Time Frame: During the first 2 cycles of treatment (1 cycle = 28 days))
NCT02143635	Drug: HDM201 Drug: ancillary treatment	Phase 1	Completed	Advanced Solid and Hematological TP53wt Tumors	Incidence of dose limiting toxicities (DLTs) (Time Frame: up to 28 days)
NCT03449381	Drug: BI 907828	Phase 1	Recruiting	Neoplasms	Phase Ia- Maximum tolerated dose (MTD) based on number of patients with dose limiting toxicities (DLTs) during first treatment cycle (Time Frame: Up to 28 days) Progression-free survival (Time Frame: Up to 24 months) Phase Ia—Number of patients with DLTs during first treatment cycle (21 days, Arm A; 28 days, Arm B) (Time Frame: Up to 28 days) Phase Ib—Number of patients with DLTs during the first treatment cycle (Time Frame: Up to 28 days)
NCT03964233	Drug: BI 907828 Drug: BI 754091 Drug: BI 754111	Phase 1	Recruiting	Neoplasms	Phase Ia—maximum tolerated dose (MTD) of BI 907828 in combination with BI 754091 based on the number of patients with DLTs during the first treatment cycle (Time Frame: Up to 21 Days) Phase Ib—Objective response (OR) (Time Frame: Up to 24 months)
NCT01664000	Drug: thioureidobutyronitrile	Phase 1	Completed	Solid Tumors	Maximum Tolerated Dose (MTD) of Kevetrin (Time Frame: Up to 6 months) Dose Limiting Toxicities (DLT) of Kevetrin. (Time Frame: up to 4 weeks)
NCT03611868	Drug: APG-115+Pembrolizumab	Phase 1 Phase 2	Recruiting	Unresectable or Metastatic Melanoma or Advanced Solid Tumors P53 Mutation MDM2 Gene Mutation	Maximum Tolerated Dose (Time Frame: 21 days) Recommended Phase II Dose (Time Frame: 21 days) Overall Response Rate (Time Frame: Up to 12 months)
NCT02264613	Drug: ALRN-6924	Phase 1 Phase 2	Completed	Solid Tumor	Evaluate the safety and tolerability of ALRN-6924 in adult patients with advanced solid tumors or lymphomas with wild-type (WT) TP53 who are refractory to or intolerant of standard therapy, or for whom no standard therapy exists—Phase 1 (Time Frame: From Day 1 of treatment until 30 days after the last cycle of treatment (each cycle is 28 days)) Evaluate the safety and tolerability of ALRN-6924 in adult patients with advanced solid tumors or lymphomas with wild-type (WT) TP53 who are refractory to or intolerant of standard therapy, or for whom no standard therapy exists—Phase 2 (Time Frame: From Day 1 of treatment until 30 days after the last cycle of treatment (each cycle is 28 days)) Determine the maximum tolerated dose (MTD)—Phase 1 (Time Frame: From the first dose until the end of the first cycle (each cycle is 28 days)) Determine Overall Response Rate—Phase 2 (Time Frame: From the first dose until the first documented date of progression or date of death from any cause, whichever comes first, assessed up to 100 months)
NCT04589845	Drug: Entrectinib Drug: Alectinib Drug: Atezolizumab Drug: Ipatasertib Drug: Trastuzumab emtansine Drug: Idasanutlin Drug: Inavolisib Drug: Belvarafenib Drug: Pralsetinib	Phase 2	Recruiting	Solid Tumor	All Cohorts: Independent Review Committee (IRC)-assessed objective response rate (ORR) based on confirmed objective response (OR) per Response Evaluation Criteria in Solid Tumors, Version 1.1 (RECIST v1.1) (Time Frame: Approximately up to 12 years)

CTCAE; Common Terminology Criteria for Adverse Events, DLT; Dose limiting toxicity, MTD; maximum tolerated dose, RECIST; The revised RECIST guidelines (versions 1.1) are available here for free with permission from the European Journal of Cancer (EJC). The guidelines and accompanying articles were published in a special issue of EJC in Jan. 2009. Most drugs as therapeutics are for the solid tumors; however, they might be adapted to include Pancreatic cancers.

**Table 2 cancers-13-03920-t002:** Recent clinical trials of PARP inhibitors in human pancreatic cancers.

Trial ID	Therapeutic Drug	Phase	Status	Condition	Primary Outcome
NCT04673448	Biological: Dostarlimab Drug: Niraparib	Phase 1	Not yet recruiting	Metastatic Pancreatic Carcinoma	Best objective response (Time Frame: 5 years)
NCT00892736	Other: Laboratory Biomarker Analysis Other: Pharmacological Study Drug: Veliparib	Phase 1	Completed	Pancreatic Carcinoma	MTD, DLT, recommended phase II dose of chronically dosed single-agent veliparib in patients with either a refractory BRCA 1/2- mutated solid cancer; platinum- refractory ovarian, fallopian tube, or primary peritoneal cancer; or basal-like breast cancer (Time Frame: 28 days)
NCT04644068	Drug: AZD5305 Drug: Paclitaxel Drug: Carboplatin	Phase 1	Recruiting	Pancreatic Cancer	The number of subjects with adverse events/serious adverse events (Time Frame: From time of Informed Consent to 28 days post last dose (approximately 1 year)) The number of subjects with dose-limiting toxicity (DLT), as defined in the protocol. (Time Frame: From first dose of study treatment until the end of Cycle 1. Approximately 35 days.)
NCT00576654	Drug: Irinotecan Hydrochloride Other: Laboratory Biomarker Analysis Other: Pharmacological Study Drug: Veliparib	Phase 1	Active, not recruiting	Stage III Pancreatic Cancer AJCC v6 and v7	Optimal biologic dose (OBD) (Time Frame: Up to day 9 of course 1) Maximum administered dose of study drugs (Time Frame: Up to 21 days) Maximally tolerated dose (MTD) of study drugs (Time Frame: Up to 21 days) Recommended phase II dose (RP2D) of study drugs (Time Frame: Up to 21 days)
NCT00047307	Drug: alvocidib Drug: gemcitabine hydrochloride Radiation: 3-dimensional conformal radiation therapy Other: laboratory biomarker analysis	Phase 1	Completed	Adenocarcinoma of the Pancreas Recurrent Pancreatic Cancer Stage II Pancreatic Cancer Stage III Pancreatic Cancer Stage IV Pancreatic Cancer	Maximum tolerated dose of flavopiridol when administered biweekly in conjunction with radiation for patients with locally advanced pancreatic or extrahepatic bile duct cancer (Time Frame: 6 weeks)
NCT04764084	Drug: Niraparib Drug: Anlotinib	Phase 1	Not yet recruiting	Pancreatic Cancer	Dose limiting toxicity (DLT) and maximum tolerated dose (MTD) (Time Frame: 4 weeks)
NCT04182516	Drug: NMS-03305293	Phase 1	Recruiting	Advanced/Metastatic Solid Tumors	Number of Participants with first-cycle dose limiting toxicity (Time Frame: Time interval between the date of the first dose administration in Cycle 1 (each cycle is 28 days) and the date of the first dose administration in Cycle 2 which is expected to be 28 days or up to 42 days in case of dose delay due to toxicity)
NCT00515866	Drug: KU-0059436 (AZD2281)(PARP inhibitor) Drug: Gemcitabine	Phase 1	Completed	Pancreatic Neoplasms	To establish the maximum tolerated dose (MTD) or a tolerable and effective dose of KU 0059436 in combination with gemcitabine (Time Frame: assessed at each visit)
NCT04503265	Drug: AMXI-5001:Dose Escalation Phase I Drug: AMXI-5001:Dose Expansion Phase II	Phase 1 Phase 2	Recruiting	Pancreatic Cancer	Determine the Maximum Tolerated Dose (MTD) (Time Frame: Approximately 12 months)
NCT01489865	Drug: ABT-888 and mFOLFOX-6	Phase 1 Phase 2	Unknown	Metastatic Pancreatic Cancer	Dose limiting toxicities (Time Frame: 28 days)
NCT03337087	Drug: Fluorouracil Other: Laboratory Biomarker Analysis Drug: Leucovorin Calcium Drug: Liposomal Irinotecan Drug: Rucaparib	Phase 1 Phase 2	Recruiting	Metastatic Pancreatic Adenocarcinoma Stage IV Pancreatic Cancer AJCC v6 and v7	Number of participants with dose limiting toxicities (Phase I) (Time Frame: Up to 28 days from start of treatment) Objective response (Phase Ib) (Time Frame: Baseline up to 3 years) Best response rate (Phase II) (Time Frame: At 32 weeks)
NCT04228601	Drug: Fluzoparib Drug: Fluzoparib placebo Drug: mFOLFIRINOX	Phase 1 Phase 2	Recruiting	Advanced Pancreatic Cancer	Number of Participants With a Dose Limited Toxicity (Time Frame: Within 28 Days after The First Dose) Maximum Tolerated Dose (Time Frame: Time Frame: Up to 8 months) Objective Response Rate (Time Frame: From Week 9 until documented disease progression or study discontinuation (approximately up to 24 months))
NCT03404960	Drug: Niraparib + Nivolumab Drug: Niraparib + Ipilimumab	Phase 1 Phase 2	Recruiting	Pancreatic Adenocarcinoma	Progression-free survival (Time Frame: 6 months after initiation of study therapy)
NCT02042378	Drug: Rucaparib	Phase 2	Completed	Pancreatic Cancer Pancreatic Ductal Adenocarcinoma	Overall Response Rate (ORR) per RECIST v1.1 as assessed by the investigator (Time Frame: Screening, within 7 days prior to the start of every 3rd cycle of treatment, and Treatment Discontinuation Visit. Study to last for ~3 years.)
NCT03682289	Drug: ATR Kinase Inhibitor AZD6738 Drug: Olaparib	Phase 2	Recruiting	Metastatic Pancreatic Cancer Stage III Pancreatic Cancer Stage IV Pancreatic Cancer	Objective response rate (ORR) (Time Frame: Up to 2.5 years) Objective response rate (ORR) for other solid tumors (Time Frame: Up to 2.5 years)
NCT04493060	Biological: Dostarlimab Drug: Niraparib	Phase 2	Recruiting	Metastatic Pancreatic Ductal Adenocarcinoma Stage IV Pancreatic Cancer AJCC v8	Disease control rate at 12 weeks (DCR12) (Time Frame: At 12 weeks)
NCT02498613	Other: 18F-Fluoromisonidazole Drug: Cediranib Maleate Other: Laboratory Biomarker Analysis Drug: Olaparib Procedure: Positron Emission Tomography	Phase 2	Recruiting	Metastatic Pancreatic Adenocarcinoma Pancreatic Ductal Adenocarcinoma Stage III Pancreatic Cancer AJCC v6 and v7 Stage IV Pancreatic Cancer AJCC v6 and v7 Unresectable Pancreatic Adenocarcinoma Unresectable Pancreatic Carcinoma	Objective response rate (Time Frame: Up to 4 weeks after completion of study treatment)
NCT04171700	Drug: Rucaparib	Phase 2	Recruiting	Solid Tumor	Best Overall Response Rate by Investigator (Time Frame: From first dose of study drug until disease progression (up to approximately 2 years))
NCT04550494	Procedure: Biopsy Drug: Talazoparib	Phase 2	Recruiting	Advanced Pancreatic Carcinoma Metastatic Pancreatic Carcinoma Stage II Pancreatic Cancer AJCC v8 Stage IIA Pancreatic Cancer AJCC v8 Stage IIB Pancreatic Cancer AJCC v8 Stage III Pancreatic Cancer AJCC v8 Stage IV Pancreatic Cancer AJCC v8	Percent of patients who demonstrate simultaneous Rad51 activation (Time Frame: At cycle 2 day 1)
NCT01286987	Drug: Talazoparib	Phase 2	Active, not recruiting	Metastatic Pancreatic Adenocarcinoma Pancreatic Ductal Adenocarcinoma Stage IV Pancreatic Cancer AJCC v6 and v7	Objective response rate (defined as complete response or partial response) assessed using Response Evaluation Criteria in Solid Tumors 1.1 (Time Frame: At 24 weeks)
NCT03601923	Drug: Niraparib	Phase 2	Recruiting	Pancreatic Cancer	Progression Free Survival (Time Frame: 6 months)
NCT04409002	Drug: Niraparib Drug: Dostarlimab Radiation: Radiation	Phase 2	Recruiting	Pancreatic Cancer Metastatic Pancreatic Cancer	Disease control rate with RECIST 1.1 (Time Frame: 3 months up to 2 years)
NCT04548752	Drug: Olaparib Biological: Pembrolizumab	Phase 2	Recruiting	Metastatic Pancreatic Adenocarcinoma Stage IV Pancreatic Cancer AJCC v8	Progression-free survival (PFS) (Time Frame: Up to 3 years)
NCT03140670	Drug: RUCAPARIB	Phase 2	Active, not recruiting	Pancreatic Cancer	Number of Adverse Events (Time Frame: 4 years)
NCT04858334	Drug: Olaparib Drug: Placebo Administration	Phase 2	Recruiting	Pancreatic Cancer	Improvement in relapse-free survival (RFS) (Time Frame: From randomization to first documentation of disease recurrence (primary tumor relapse) or death, assessed from 22 months to 44 months)
NCT02184195	Drug: Olaparib Drug: Placebo	Phase 3	Active, not recruiting	Germline BRCA1/2 Mutations and Metastatic Adenocarcinoma of the Pancreas	Progression-free Survival (PFS) by Blinded Independent Central Review (BICR) Using Modified Response Evaluation Criteria in Solid Tumours. This Study Used Modified RECIST Version (v) 1.1 (RECIST v1.1) (Time Frame: Up to 4 years)

DLT; Dose limiting toxicity, MTD: maximum tolerated dose, RECIST; The revised RECIST guidelines (versions 1.1) are available here for free with permission from the European Journal of Cancer (EJC). The guidelines and accompanying articles were published in a special issue of EJC in January 2009.

**Table 3 cancers-13-03920-t003:** Recent clinical trials based on gene therapy in human pancreatic cancers.

Trial ID	Therapeutic Drug	Phase	Status	Condition	Primary Outcome
NCT00711997	Biological: DTA-H19	Phase 1 Phase 2	Completed	Pancreatic Neoplasms	Maximal Tolerated Dose (MTD) & Dose Limiting Toxicity (DLT) of Intratumoral Injections of BC-819 (Time Frame: Week 4)
NCT00769483	Drug: MK-0646 Drug: Gemcitabine Drug: Erlotinib	Phase 1 Phase 2	Completed	Pancreatic Cancer Pancreatic Adenocarcinoma	MK-0646 Maximum Tolerable Dose (Time Frame: Up to 12 cycles) Progression Free Survival (Time Frame: From date of randomization until the date of first documented progression or date of death from any cause, whichever came first, assessed up to 100 months)
NCT03602079	Drug: A166	Phase 1 Phase 2	Recruiting	Pancreatic Cancer	Phase I: Maximum Tolerated Dose (Time Frame: Minimum of 21 days from date of enrollment until the date of first documented progression or date of death from any cause, whichever came first, assessed up to 24 months) Phase II: Percentage of patients with an Objective Response Rate (ORR) (Complete Response (CR) + Partial Response (PR)) (Time Frame: From date of enrollment until the date of first documented progression or date of death from any cause, whichever came first, assessed up to 24 months)
NCT03190941	Drug: Cyclophosphamide Drug: Fludarabine Biological: Anti-KRAS G12V mTCR PBL Drug: Aldesleukin	Phase 1 Phase 2	Suspended (Administratively Suspended)	Pancreatic Cancer	Response rate (Time Frame: 6 weeks and 12 weeks following administration of the cell product, then every 3 months × 3, then every 6 months × 2 years, then per PI discretion) Frequency and severity of treatment-related adverse events (Time Frame: From time of cell infusion to two weeks after cell infusion)
NCT02705196	Genetic: delolimogene mupadenorepvec Drug: gemcitabine Drug: nab-paclitaxel Biological: atezolizumab	Phase 1 Phase 2	Recruiting	Pancreatic Cancer	Number of patient with dose-limiting toxicities (DLTs) as evaluated accordingly to CTCAE 4.0 (Time Frame: 9 months)
NCT03745326	Drug: Cyclophosphamide Drug: Fludarabine Drug: Aldesleukin Biological: anti-KRAS G12D mTCR PBL	Phase 1 Phase 2	Suspended ((suspension))	Pancreatic Cancer	Frequency and severity of treatment-related adverse events (Time Frame: From time of cell infusion to two weeks after cell infusion) Response rate (Time Frame: 6 weeks and 12 weeks following administration of the cell product, then every 3 months × 3, then every 6 months × 2 years, then per PI discretion)
NCT01583686	Drug: Fludarabine Biological: Anti-mesothelin chimeric T cell receptor (CAR) transduced peripheral blood lymphocytes (PBL) Drug: Cyclophosphamide Drug: Aldesleukin	Phase 1 Phase 2	Terminated (Study terminated due to slow/insufficient accrual.)	Pancreatic Cancer	Number of Patients With Objective Tumor Regression (Time Frame: 3.5 mos.) Number of Participants With Serious and Non-serious Adverse Events Assessed by the Common Terminology Criteria in Adverse Events (CTCAE v4.0) (Time Frame: Date treatment consent signed to date off study, approximately 6 months and 17 days for Group A01, 16 months and 13 days for Group A02, 13 months and 3 days for Group A03, 10 months and 16 days for Group A04, and 11 months and 26 days for Group A05. )
NCT03192462	Biological: multiTAA specific T cells	Phase 1 Phase 2	Active, not recruiting	Pancreatic Cancer	Number of patients with treatment related serious adverse events (Time Frame: 7 months) Number of patients who received 6 infusions of multiTAA-specific T cells (Time Frame: 6 months)
NCT02830724	Drug: Cyclophosphamide Drug: Fludarabine Drug: Aldesleukin Biological: Anti-hCD70 CAR transduced PBL	Phase 1 Phase 2	Suspended ((suspension))	Pancreatic Cancer	Frequency and severity of treatment-related adverse events (Time Frame: From time of cell infusion to two weeks after cell infusion) Response rate (Time Frame: 6 weeks and 12 weeks following administration of the cell product, then every 3 months × 3, then every 6 months × 2 years, then per PI discretion)
NCT00255827	Biological: HyperAcute-Pancreatic Cancer Vaccine	Phase 1 Phase 2	Completed	Pancreatic Cancer	To assess the side effects, dose-limiting toxicity and maximum tolerated dose. (Time Frame: 6 months)
NCT04637698	Biological: OH2 injection	Phase 1 Phase 2	Recruiting	Pancreatic Cancer	The objective response rate of patients with pancreatic cancer receiving OH2 injection. (Time Frame: 2 years)
NCT00959946	Drug: Bosutinib Drug: Capecitabine	Phase 1 Phase 2	Terminated	Advanced Breast Cancer (Parts 1 and 2) Advanced Pancreatic Cancer (Part 1) Advanced Colorectal Cancer (Part 1) Advanced Cholangiocarcinoma (Part 1) Advanced Glioblastoma Multiforme (Part 1)	Maximum Tolerated Dose (MTD)—Part 1 (Time Frame: Part 1 Baseline up to Day 21) Percentage of Participants With Treatment-Emergent Adverse Events (AEs) or Serious Adverse Events (SAEs)—Part 1 (Time Frame: Part 1 Baseline up to 28 days after last dose of study treatment)
NCT04426669	Drug: Cyclophosphamide Drug: Fludarabine Biological: Tumor-Infiltrating Lymphocytes (TIL) Drug: Aldesleukin	Phase 1 Phase 2	Recruiting	Pancreatic Cancer	Maximum tolerated dose (MTD) (Time Frame: 28 Days Post IL-2) Preliminary efficacy of tumor reactive autologous lymphocytes with knockout of CISH gene in patients with refractory metastatic gastrointestinal epithelial cancers: changes in diameter (Time Frame: Every 4 Weeks for the first three months, then every 8 weeks thereafter, up to 2 years) Safety of tumor reactive autologous lymphocytes with knockout of the CISH gene—Incidence of Adverse Events (Time Frame: 2 Years or Disease Progression)
NCT04739046	Theragene arm	Phase 2	Recruiting	Pancreas Cancer	Objective Response Rate (Time Frame: 24 weeks)
NCT02340117	Genetic: SGT-53 Drug: nab-paclitaxel Drug: Gemcitabine	Phase 2	Recruiting	Metastatic Pancreatic Cancer	Progression free survival (PFS) at 5.5 months (Time Frame: 5.5 months) Objective response rate (ORR) (Time Frame: Up to 5 years)
NCT02806687	Drug: Gene Therapy product CYL-02 Drug: Gemcitabine	Phase 2	Recruiting	Pancreatic Adenocarcinoma	Progression-free survival (Time Frame: From date to randomization until the date of first documented progression or date of death, whichever came first, assessed up to 12 months)
NCT00868114	Biological: KLH-pulsed autologous dendritic cell vaccine	Phase 2	Terminated (Poor recruitment)	Metastatic Pancreatic Cancer	Overall Survival (Time Frame: Patients will be followed until death)
NCT00305760	Drug: Cetuximab Biological: Pancreatic tumor vaccine Drug: Cyclophosphamide	Phase 2	Completed	Pancreatic Cancer	Safety of Combining the Pancreatic Tumor Vaccine in Sequence With Cyclophosphamide and Erbitux. Safety is Defined as the Number of Treatment-related Grade 3 or 4 Adverse Events Observed in Greater Than 5% of the Patient Population (Time Frame: 7 months)
NCT00084383	Biological: GVAX pancreatic cancer vaccine	Phase 2	Completed	Pancreatic Cancer	Overall Survival (Time Frame: Participants were followed for the duration of the study, an average of 2 years)Disease-free Survival (Time Frame: Participants were followed for the duration of the study, an average of 2 years)
NCT00389610	Biological: allogenic GM-CSF plasmid-transfected pancreatic tumor cell vaccine	Phase 2	Active, not recruiting	Pancreatic Cancer	Safety as measured by local and systemic toxicities (Time Frame: Until progression)
NCT04548752	Drug: Olaparib Biological: Pembrolizumab	Phase 2	Recruiting	Metastatic Pancreatic Adenocarcinoma Stage IV Pancreatic Cancer AJCC v8	Progression-free survival (PFS) (Time Frame: Up to 3 years)
NCT00305760	Drug: Cetuximab Biological: Pancreatic tumor vaccine Drug: Cyclophosphamide	Phase 2	Completed	Pancreatic Cancer	Safety of Combining the Pancreatic Tumor Vaccine in Sequence With Cyclophosphamide and Erbitux. Safety is Defined as the Number of Treatment-related Grade 3 or 4 Adverse Events Observed in Greater Than 5% of the Patient Population (Time Frame: 7 months)
NCT00084383	Biological: GVAX pancreatic cancer vaccine	Phase 2	Completed	Pancreatic Cancer	Overall Survival (Time Frame: Participants were followed for the duration of the study, an average of 2 years) Disease-free Survival (Time Frame: Participants were followed for the duration of the study, an average of 2 years)
NCT00389610	Biological: allogenic GM-CSF plasmid-transfected pancreatic tumor cell vaccine	Phase 2	Active, not recruiting	Pancreatic Cancer	Safety as measured by local and systemic toxicities (Time Frame: Until progression)
NCT04548752	Drug: Olaparib Biological: Pembrolizumab	Phase 2	Recruiting	Metastatic Pancreatic Adenocarcinoma Stage IV Pancreatic Cancer AJCC v8	Progression-free survival (PFS) (Time Frame: Up to 3 years)
NCT01088789	Biological: PANC 10.05 pcDNA-1/GM-Neo and PANC 6.03 pcDNA-1 neo vaccine.	Phase 2	Recruiting	Pancreatic Cancer	Disease free overall survival. (Time Frame: Total of 13 years with 6 months between vaccines)
NCT04550494	Procedure: Biopsy Drug: Talazoparib	Phase 2	Recruiting	Advanced Pancreatic Carcinoma Metastatic Pancreatic Carcinoma Stage II Pancreatic Cancer AJCC v8 Stage IIA Pancreatic Cancer AJCC v8 Stage IIB Pancreatic Cancer AJCC v8 Stage III Pancreatic Cancer AJCC v8 Stage IV Pancreatic Cancer AJCC v8	Percent of patients who demonstrate simultaneous Rad51 activation (Time Frame: At cycle 2 day 1)
NCT04383210	Drug: Seribantumab	Phase 2	Recruiting	Pancreatic Cancer	Objective Response Rate (Time Frame: Up to 12 months)
NCT04171700	Drug: Rucaparib	Phase 2	Recruiting	Solid Tumor	Best Overall Response Rate by Investigator (Time Frame: From first dose of study drug until disease progression (up to approximately 2 years))
NCT02405585	Drug: mFOLFIRINOX Biological: Algenpantucel-L Immunotherapy Radiation: SBRT Drug: Gemcitabine	Phase 2	Terminated	Pancreatic Cancer Pancreatic Carcinoma Non-resectable	Progression Free Survival (Time Frame: 18 months (assuming enrollment period of 1 year))
NCT00727441	Biological: GVAX pancreatic cancer vaccine Drug: cyclophosphamide	Phase 2	Completed	Pancreatic Cancer	Safety as Measured by Number of Participants With Treatment-related Grade 3 or 4 Local and Systemic Toxicity as Defined by NCI CTCAE v3.0 (Time Frame: 7 years) Amount of T-regulatory Cells (Tregs) and CD4+ and CD8+ Effector T Cells, After Neoadjuvant GVAX Pancreatic Cancer Vaccination. (Time Frame: Up to 8 years) Change in the Number and Function of Peripheral Mesothelin-specific CD8+ T Cells and CD4+, FoxP3+, and GITR+ Tregs (Time Frame: Up to 8 years)
NCT00051467	Genetic: TNFerade	Phase 3	Completed	Pancreatic Cancer	Not Provided
NCT01836432	Drug: FOLFIRINOX Biological: Algenpantucel-L Immunotherapy Radiation: 5-FU Chemoradiation Drug: Gemcitabine Drug: Capecitabine Drug: Nab-Paclitaxel	Phase 3	Terminated (Company decision)	Pancreatic Cancer Pancreatic Carcinoma Non-resectable Locally Advanced Malignant Neoplasm	Overall Survival (Time Frame: 13.5 months (assuming enrollment period of 1–2 years))
NCT01286987	Drug: Talazoparib	Phase 2	Active, not recruiting	Metastatic Pancreatic Adenocarcinoma Pancreatic Ductal Adenocarcinoma Stage IV Pancreatic Cancer AJCC v6 and v7	Objective response rate (defined as complete response or partial response) assessed using Response Evaluation Criteria in Solid Tumors 1.1 (Time Frame: At 24 weeks)

CISH; Cytokine inducible SH2 containing protein, CTCAE; Common Terminology Criteria for Adverse Events (CTAE), DLT; Dose limiting toxicity, GVAX; a granulocyte-macrophage colony-stimulating factor (GM-CSF) gene -transfected tumor cell vaccine, MTD: maximum tolerated dose, OH2; a novel oncolytic virus derived from the wild type HSV-2 strain HG52, Panc 10.05:a pancreatic adenocarcinoma epithelial cell line derived in 1992 from a primary tumor removed from the head of the pancreas of a male with pancreatic adenocarcinoma.

**Table 4 cancers-13-03920-t004:** Recent clinical trials based on immunotherapy in human pancreatic cancers.

Trial ID	Therapeutic Drug	Phase	Status	Condition	Primary Outcome
NCT00603863	Biological: IMMU-107 (hPAM4)	Phase 1 Phase 2	Completed	Pancreatic Cancer	Safety will be evaluated based upon physical examinations, hematology, and chemistry laboratory testing as well as toxicity (Time Frame: Over 12 weeks)
NCT01631552	Drug: Sacituzumab Govitecan (SG)	Phase 1 Phase 2	Completed	Pancreatic Cancer	Percentage of Participants Experiencing Any Treatment Emergent Adverse Events and Serious Treatment Emergent Adverse Events (Time Frame: First dose date up to last dose (maximum duration: 55.2 months) plus 30 days Percentage of Participants Who Permanently Discontinued Sacituzumab Govitecan (SG) Due to Any Adverse Events, Excluding Adverse Events Leading to Death (Time Frame: First dose date up to last dose (maximum duration: 55.2 months)) Percentage of Participants Who Required Dose Interruption Due to Any Adverse Events (Time Frame: First dose date up to last dose (maximum duration: 55.2 months)) Objective Response Rate (ORR) by Independent Central Review (ICR) (Time Frame: Up to 74 months) Objective Response Rate by Local Assessment (Time Frame: Up to 74 months)
NCT01956812	Drug: IMMU-107 Drug: placebo Drug: Gemcitabine	Phase 3	Terminated (The DSMB conducted an interim analysis on overall survival, which showed that the treatment arm did not demonstrate a sufficient improvement in OS vs. placebo.)	Metastatic Pancreatic Cancer Pancreatic Cancer	overall survival (Time Frame: 24 months)

DSMB; Data Safety Monitoring Board.

**Table 5 cancers-13-03920-t005:** Models of the multifunctionality of mutant p53. Mut-p53 induces various biological functions, such as chronic inflammation, deregulation of cellular metabolism, high proliferation rate, genomic instability, metastasis, resistance to apoptosis/therapeutic drugs, and stem cells. The figure is adapted and modified to show the various biological functions induced by mutp53 [38].

Mutant p35						
Metastasis	High Proliferation Rate	Resistance to Apoptosis/Drugs	Stem Cell	Chronic Inflammation	Genomic Instability	Metabolism
CDH1	c-MYC	MDR1	CD133	sIL-1Ra	BRCA1	SREBP
TFG-β	NF-Y	VDR	OCT3/4	TLR3	E2F	RhoA/ROCK
MAP2K1	MAP2K3	IGF2	CD44	NF-κB	RAD17	ETS2
SENP1/RAC1	CDK1	p63	NANOG	HIF1α	SP1	GLUT1
TWIST1	IGF1R	p71	SOX2	STAT3	MRE11	HK2

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
