# Peer review of "Therapeutic Strategies Targeting Tumor Suppressor Genes in Pancreatic Cancer"

_cancers, 2021, doi:10.3390/cancers13153920_

Round 1

Reviewer 1 Report

Dr. Kuo and the colleagues summarized the role of tumor suppresser genes (TSGs) and the therapeutic targets in pancreatic ductal adenocarcinoma (PDAC) in this manuscript. Information of clinical trials could be especially useful, but some revisions should be required for publication, as follows:

- 2.1. Olaparib: This paragraph is composed of just 1 sentence. The authors might include the one sentence to the above paragraph and change the title of 2. and 2.1.

- 2.2.: There is no sentence, juts only Table 1. Elucidate Table 1 and call out “(Table 1)” in the text like (Figure 1).

- 3.1. : Delete period “.” before Gain-of-function.

- Table 1. NCT03755739, condition: The authors should delete the descriptions (hepatocellular carcinoma, Lung cancer etc.) except pancreatic cancer.

- 7. “Cancer immunotherapy”: Looks like a general title. Alter the title to more specific and appropriate one in this review article. Please double check all titles.

- 7.1.: Is there any TSG information?

- 7.3. and 7.4.: Change the titles as well.

- p14, last line: Edit “Trp53Lox/++” to “Trp53Lox/+” (delete one plus).

The manuscript should be revised by a native speaker finally.

Author Response

[Reviewer 1]

Comments and Suggestions for Authors

Dr. Kuo and the colleagues summarized the role of tumor suppresser genes (TSGs) and the therapeutic targets in pancreatic ductal adenocarcinoma (PDAC) in this manuscript. Information of clinical trials could be especially useful, but some revisions should be required for publication, as follows:

- 2.1. Olaparib: This paragraph is composed of just 1 sentence. The authors might include the one sentence to the above paragraph and change the title of 2. and 2.1.

[Answer]

As suggested by the reviewer, this section was moved to the previous paragraph. Moreover, we changed the title of sections 2 and 2.1.

- 2.2.: There is no sentence, juts only Table 1. Elucidate Table 1 and call out “(Table 1)” in the text like (Figure 1).

[Answer]

As suggested by the reviewer, we have added text in this subsection of the manuscript.

- 3.1. : Delete period “.” before Gain-of-function.

[Answer]

As suggested by the reviewer, we have deleted this period.  

- Table 1. NCT03755739, condition: The authors should delete the descriptions (hepatocellular carcinoma, Lung cancer etc.) except pancreatic cancer.

[Answer]

As suggested by the reviewer, we have deleted the descriptions of all cancers other than pancreatic cancer.

- 7. “Cancer immunotherapy”: Looks like a general title. Alter the title to more specific and appropriate one in this review article. Please double check all titles.

[Answer]

As suggested by the reviewer, we have replaced the titles with a more specific and appropriate ones.

- 7.1.: Is there any TSG information?

[Answer]

As suggested by the reviewer, we added information on TSGs.

- 7.3. and 7.4.: Change the titles as well.

[Answer]

As suggested by the reviewer, we have changed the headlines of these subsections.

- p14, last line: Edit “Trp53Lox/++” to “Trp53Lox/+” (delete one plus).

[Answer]

As suggested by the reviewer, we have corrected this word.

The manuscript should be revised by a native speaker finally.

[Answer]

As suggested by the reviewer, the manuscript has been reviewed by an experiences editor whose first language is English.

Reviewer 2 Report

This is an excellent review. I have no major or minor issues: it can be bublished after minor review of the English language  

Author Response

[Reviewer 2]

This is an excellent review. I have no major or minor issues: it can be published after minor review of the English language  

[Answer]

As suggested by the reviewer, the manuscript has been reviewed by an experiences editor whose first language is English.

Reviewer 3 Report

This is an extensive review about possibilities of targeted therapy in pancreatic cancer. The AA first describe genetic alterations most commonly observed in PDAC, its consequences in carcinogenesis, and then proceed to possible translation into therapeutics. It is well written and references are extensive as well.

Author Response

[Reviewer 3]

This is an extensive review about possibilities of targeted therapy in pancreatic cancer. The AA first describe genetic alterations most commonly observed in PDAC, its consequences in carcinogenesis, and then proceed to possible translation into therapeutics. It is well written and references are extensive as well.

[Answer]

We thank the reviewer for these comments.

Reviewer 4 Report

In this manuscript, Kuo and colleagues collect a number of evidences and reports clearly indicating the urgent need and significance of better therapeutic strategies targeting tumor suppressor gene products to prevent the progression of pancreatic cancer.  

The review discuss a number of important studies and several recent clinical trials, which can be considered very relevant for both pre-clinical researchers and clinicians.

However, the manuscript requires adequate editing in order to deliver the message to the reader. In fact, the main problem of these "collections" is that, sometimes, the amount of information reported can result overwhelming.

Specific comments:

Introduction requires editing: It seems to me that the authors provided few times (back to back) a description of pancreatic cancer and then of genes and therapies. This need to be harmonized.

Table 1 looks overcrowded, and create confusions, this need to be modified, better divide the table into two either on the basis of Phase or current status.

Authors should work on consistency, some headings are bold, and subheadings are normal, headings under subheadings bold.  This need to be harmonized to make it easily understandable and more attractive to the readers.

Figure 4 is more of a table.

The manuscript requires fine editing for its style. Like Page 10 paragraph 2, CP-31398, PRIMA-1 etc bold at some places and not bold at other places, should be evaluated more carefully.

Page 11, Line 3-6: unusual font size

Page 11, Paragraph 1: fragmented sentence

Page 11, Paragraph 1, line 3-4 reference missing

Page 13, Paragraph 1, inhibitors name bold at some places and not bold at other places

Page 13, Paragraph 2, line 1-2 reference missing

Authors should discuss and comment on results obtained from different studies and present them in details to conclude the article and make it easily understandable and more attractive to the readers.   

Author Response

[Reviewer 4]

In this manuscript, Kuo and colleagues collect a number of evidences and reports clearly indicating the urgent need and significance of better therapeutic strategies targeting tumor suppressor gene products to prevent the progression of pancreatic cancer.  

The review discuss a number of important studies and several recent clinical trials, which can be considered very relevant for both pre-clinical researchers and clinicians.

However, the manuscript requires adequate editing in order to deliver the message to the reader. In fact, the main problem of these "collections" is that, sometimes, the amount of information reported can result overwhelming.

 [Answer]

As suggested by the reviewer, the manuscript has been reviewed by an experiences editor whose first language is English. We also shortened the original manuscript and relevant references. 

Specific comments:

*Introduction requires editing: It seems to me that the authors provided few times (back to back) a description of pancreatic cancer and then of genes and therapies. This need to be harmonized.

[Answer]

As suggested by the reviewer, we have reorganized the stream of content of the manuscript to describe pancreatic cancer, followed by pertinent molecules and therapies.  

*Table 1 looks overcrowded, and create confusions, this need to be modified, better divide the table into two either on the basis of Phase or current status.

[Answer]

As suggested by the reviewer, we have divided the Tables into four tables according to the agendas of tumor suppressor proteins and recent therapies. We have deleted older published therapies.

*Authors should work on consistency, some headings are bold, and subheadings are normal, headings under subheadings bold.  This need to be harmonized to make it easily understandable and more attractive to the readers.

[Answer]

We apologize for the lack of consistency in formatting of the manuscript. As suggested by the reviewer, we have corrected this issue.

*Figure 4 is more of a table.

[Answer]

We replaced Figure 4 with Table V.

*The manuscript requires fine editing for its style. Like Page 10 paragraph 2, CP-31398, PRIMA-1 etc bold at some places and not bold at other places, should be evaluated more carefully.

[Answer]

As suggested by the reviewer, we have edited and corrected these inconsistencies.

*Page 11, Line 3-6: unusual font size

[Answer]

As suggested by the reviewer, we have corrected this issue.

*Page 11, Paragraph 1: fragmented sentence

[Answer]

As suggested by the reviewer, we have revised this sentence.

*Page 11, Paragraph 1, line 3-4 reference missing

[Answer]

As suggested by the reviewer, we have added the missing references.  

*Page 13, Paragraph 1, inhibitors name bold at some places and not bold at other places

[Answer]

As suggested by the reviewer, we have corrected these inconsistencies.

*Page 13, Paragraph 2, line 1-2 reference missing

[Answer]

As suggested by the reviewer, we have added the missing references.  

*Authors should discuss and comment on results obtained from different studies and present them in details to conclude the article and make it easily understandable and more attractive to the readers.

[Answer]

As suggested by the reviewer, we have added the results of different studies, commented on each section, and inserted a conclusion.

Round 2

Reviewer 1 Report

The authors have revised according to my suggestions. The revised version of the manuscript could be acceptable for publication after final check.